# Canadian Continental-Scale Hydrology under a Changing Climate: A Review

Tricia A. Stadnyk [1,*] and Stephen J. Déry [2]

1 Department of Geography, Faculty of Arts, University of Calgary, Calgary, AB T2N 1N4, Canada
2 Department of Geography, Earth and Environmental Sciences, Faculty of Environment, University of Northern British Columbia, Prince George, BC V2N 4Z9, Canada; Stephen.Dery@unbc.ca
* Correspondence: tricia.stadnyk@ucalgary.ca; Tel.: +1-403-220-6586

**Abstract:** Canada, like other high latitude cold regions on Earth, is experiencing some of the most accelerated and intense warming resulting from global climate change. In the northern regions, Arctic amplification has resulted in warming two to three times greater than global mean temperature trends. Unprecedented warming is matched by intensification of wet and dry regions and hydroclimatic cycles, which is altering the spatial and seasonal distribution of surface waters in Canada. Diagnosing and tracking hydrologic change across Canada requires the implementation of continental-scale prediction models owing the size of Canada's drainage basins, their distribution across multiple eco- and climatic zones, and the scarcity and paucity of observational networks. This review examines the current state of continental-scale climate change across Canada and the anticipated impacts to freshwater availability, including the role of anthropogenic regulation. The review focuses on continental and regional-scale prediction that underpins operational design and long-term resource planning and management in Canada. While there are significant process-based changes being experienced within Canadian catchments that are equally—if not more so—critical for community water availability, the focus of this review is on the cumulative effects of climate change and anthropogenic regulation for the Canadian freshwater supply.

**Keywords:** Canada; freshwater discharge; water supply; runoff; streamflow; climate change; regulation; arctic amplification

## 1. Canada's Changing Climate

The recent release of the Canadian Climate Change Report in 2019 (CCCR2019) confirmed what many Canadian scientists have been warning of for decades: climate change is here, it is very real, and it is hitting Canada harder than most other regions of the world. The Arctic region is warming at a rate 1.5 to 4.5 times faster than the global mean [1], which has significant implications for Canada as more than 35% of the global pan-Arctic basin (contributing to the Arctic Ocean) being Canadian territory [2], and >40% of Canada being classified as Arctic.

### 1.1. High Latitude Warming

Most of Canada has already experienced an average of 1.7 °C warming since the mid 1900's [3], with the most rapid warming occurring in the past two decades. Relative to the 1981–2010 baseline period, increases ranging from 1 to more than 5 °C are likely to occur by 2070 across the continental interior of Canada [4], which is at least twice as fast, and up to three times that, of the global mean temperature rise [3]. Arctic amplification implies warming is increasing disproportionately at higher latitudes, with temperatures up to 6 °C warmer than pre-industrial levels already witnessed across parts of the Arctic region [5].

Despite considerable uncertainty among future climate models and projections, there is a *high degree of confidence* that temperatures will continue to increase across Canada, and

at a rate faster than the global mean temperature. The CCCR2019 states it is *virtually certain* that Canada's climate will continue to warm over the 21st century [3].

### 1.2. Wet Gets Wetter, Dry Gets Drier

Precipitation is generally expected to increase across most of Canada, but there is considerable uncertainty regarding the magnitude and seasonality of those increases, which vary widely by region. Canada's continental interior is generally getting wetter along a west to east and south to north gradient, with increases of more than 35% above historic (1981–2010) annual means projected in the northern regions of the Hudson Bay basin, and a possible (but less certain) 5% decrease projected for some prairie basins of the Nelson River [4]. Based on the Intergovernmental Panel on Climate Change's (IPCC's) fifth assessment report, there is *high confidence* that precipitation at higher latitudes will increase, but less confidence in mean seasonal increases or decreases. The most significant seasonal changes are anticipated for winter [6]. Decreases in mean seasonal summer precipitation are projected by climate models for Canada, but with a relatively low degree of confidence (relative to projected increases) [3]. It is generally thought that, globally, wet regions are tending towards becoming wetter and dry regions are becoming drier [7]. Under such a scenario, it is possible the Canadian Prairies and Palliser's triangle (a semi-arid region of the western Canadian Prairies) may become drier, particularly if evapotranspiration outpaces smaller increases in precipitation under much warmer climates [8].

### 1.3. Hydro-Climatic Extremes

Changes to extreme events are particularly concerning: the frequency of extreme precipitation is projected to increase for much of Canada, with lower recurrence time between events [3]. The combined effect of more extreme precipitation scenarios (including drought) and higher temperatures is a projected increase in fire weather [3], with events such as the 2016 Fort McMurray, Alberta Wildfire being attributed with reasonable confidence to anthropogenic climate change [9,10]. The World Meteorological Organisation has reported an increase in the number and cost of extreme events world-wide [11], which is supported by the Insurance Bureau of Canada and a recent report by the Intact Centre on Climate Adaption [12]. The Canadian Natural Disaster Database (CDD) provides information on significant disaster events tracked by the Emergency Management Framework meeting a specific set of criteria, documenting when and where the event occurred, number of injuries, evacuations and fatalities, and the cost [13]. Data obtained from the CDD indicate that the majority of natural disasters in Canada are related to extreme precipitation (storms) and flood events (Figure 1). Flooding is estimated to be Canada's costliest natural disaster, with Canada receiving an average grade of 'C' for flood preparedness [12].

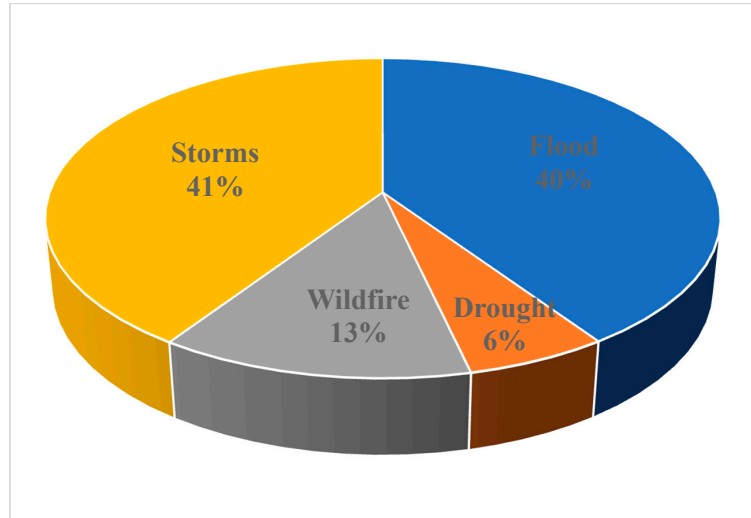

**Figure 1.** Fraction of Canadian natural disasters attributed to event types from the CDD (1900–2020) [13].

Natural disaster occurrence has notably increased in recent decades according to the CDD (Figure 2). Though inexact and incomplete records are expected to result in a lower number of natural disasters in the early 1900s, the steady increase in occurrence is still apparent and statistically significant. This leaves little doubt that increased investment in climate change preparedness, early warning and prediction systems, and adaptation and mitigation measures are critically needed for Canada.

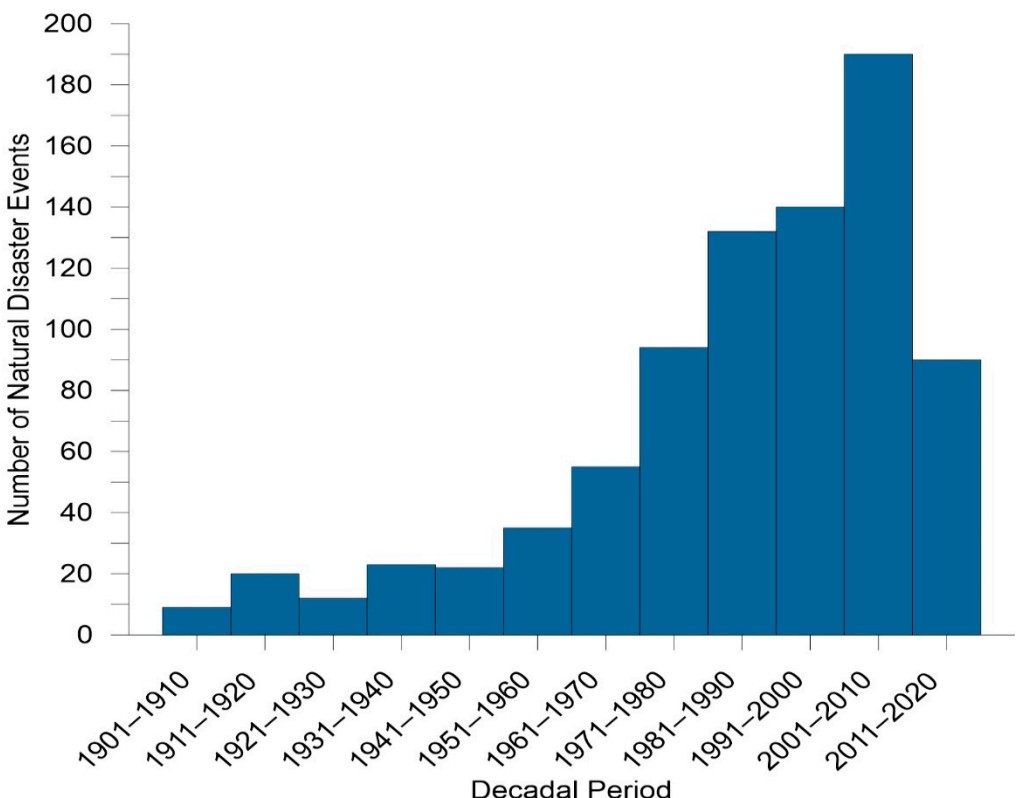

**Figure 2.** Frequency of occurrence of natural disasters across Canada by decade since the start of CDD records [13]. The most recent decadal period is incomplete, with CDD up to 2020 included in this analysis.

## 2. Assessing Changes in Freshwater Availability

The Canadian landmass is divided into six continental drainage basins contributing to three oceans: Atlantic and Gulf of Mexico, Pacific, and Arctic (including Hudson Bay). Canada's contribution to the global oceans is not insignificant at 4280 km³ per year (750 km³yr⁻¹ to the Atlantic Ocean, 2000 km³yr⁻¹ to the Arctic Ocean, and 1530 km³yr⁻¹ to Pacific Ocean), which relative to an estimated total global annual discharge of 36,000 km³/year means that Canada accounts for 11.9% of total continental freshwater discharge [14]. Continental drainage regions in Canada include the eastern Atlantic including the St. Lawrence and Great Lakes basins, the Hudson Bay interior, the Mackenzie and Arctic basins, Pacific and Yukon basins, and a small portion of the Missouri basin (Milk and St. Mary's Rivers) along the Alberta–Saskatchewan–USA borders (Figure 3). Characteristics of these basins, including their hydrometric regions and a summary of their gauging networks are presented in Table 1.

Continental-scale hydrologic and land surface models play a crucial role in discerning the quantity of freshwater supply across Canada, and therefore in quantifying the impacts of climate change. Models are specifically required due to the size of these major drainage basins, spanning multiple provinces and territories and international boundaries, and extending across multiple hydroclimatic ecozones (Table 1). Complicating even the most basic assessment of contemporary hydrology is the relatively (based on World Meteoro-

logical Organisation standards) poor spatial distribution and relatively short timeseries of observations for both meteorological and hydrometric information (Table 1). Nearly 40% of Canada's terrestrial landmass is currently ungauged and 50% is considered under-gauged [16]. Fundamental to hydrologic assessment in Canada—historic or future—is the establishment of continental-scale prediction systems forced by meteorological observations or reanalysis products, which must be rigorously evaluated against hydrologic and hydrometric observations. Assessing hydrologic change across Canada therefore necessitates a reliance on continental-scale prediction systems, particularly to map the hydrology of ungauged regions. It should not be understated that such endeavours contain significant uncertainty when input and evaluation datasets are both spatially and temporally scarce or inconsistent, adding to the complexity and uncertainty of continental prediction. We review emerging data available for model evaluation at the continental scale, and on-going research to produce hydrologic predictions for Canada's major drainage regions. Based on the Canadian Climate Change Report 2019 (CCCR2019) synthesis [3], we summarize projected changes in snow water equivalent (SWE) and streamflow in response to changes in air temperature and precipitation for the mid- to late-21st century (2020–2100; Figure 4). A qualitative assessment of agreement among various studies and projections (i.e., degree of confidence) and the amount of evidence available (i.e., robustness of projections) is provided using the IPCC's fifth assessment report (AR5) classifications based on evidence and agreement [17].

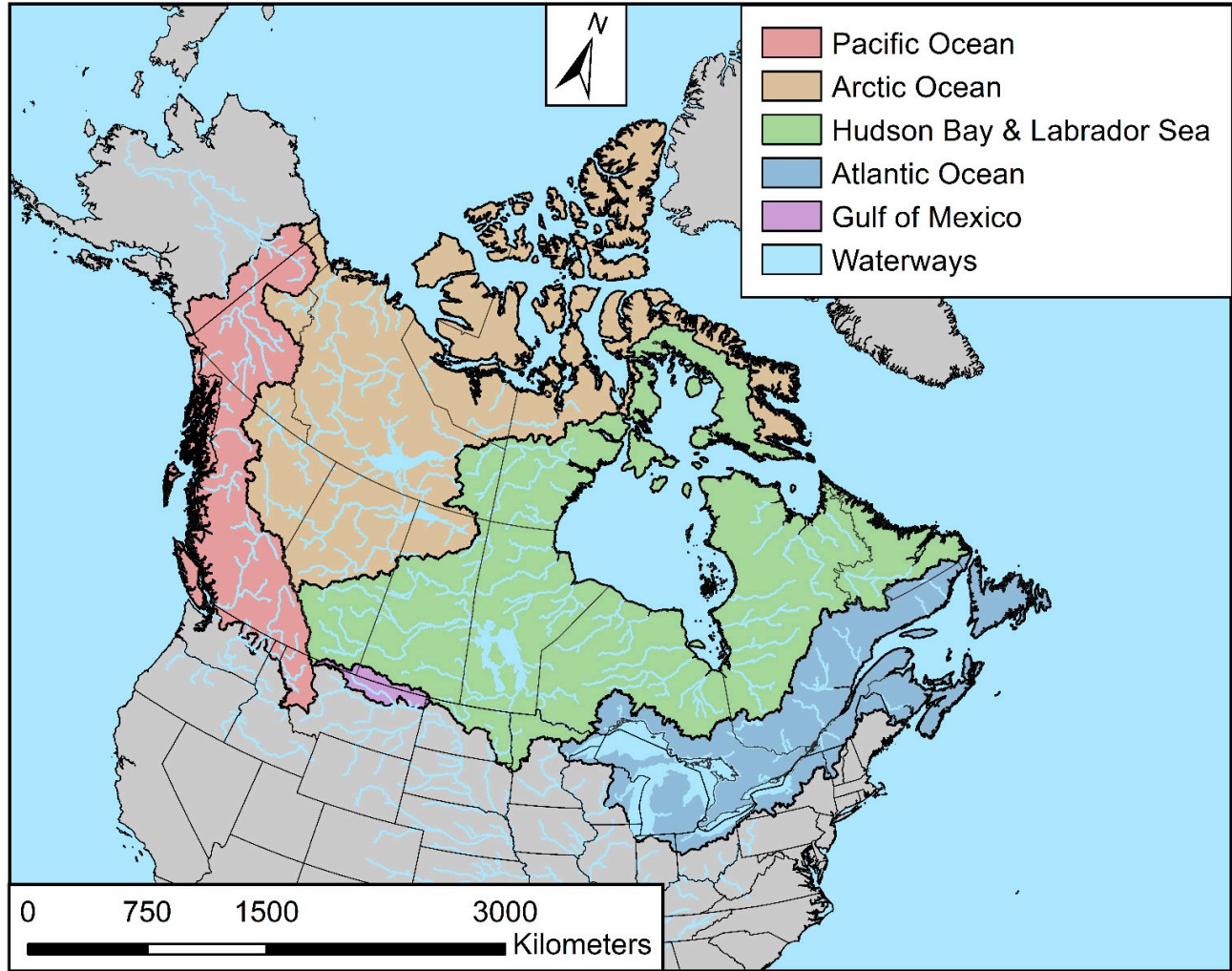

**Figure 3.** Canada's major continental drainage basins and waterways. Corresponding drainage areas and Water Survey of Canada hydrometric region codes reported in Table 1.

**Table 1.** Characteristics of Canada's major drainage basins.

| Drainage Region | Drainage Area (10⁶ km²) | Percent of Terrestrial Landmass | Water Survey of Canada Hydrometric Regions | Dominant Ecozone (Others) [1] | No. Active Gauges (% Continuous) [2] | % Regulation [3] | Hydrometric Data from–to |
|---|---|---|---|---|---|---|---|
| Atlantic/St. Lawrence-Great Lakes | 1.2 | 12.3 | 01, 02 | Dfb/Dfc (Dfa) | 948 (86%) | 46% | 1850–2021 |
| Hudson Bay/Labrador | 4.1 | 41.4 | 03, 04, 05, 06 | Dfc (BSk, Dwc, Dfa, Dfb) | 1106 (43%) | 48% | 1813–2021 |
| Missouri/Milk-St. Mary's | 0.027 | 0.3 | 11 | Dfb (BSk, Bwk) | 68 (14%) | 74% | 1908–2021 |
| Mackenzie/Arctic | 3.6 | 35.9 | 07, 10 | Dfc (Dsb, Dfb) | 390 (61%) | 19% | 1913–2021 |
| Pacific/Yukon | 1.0 | 10.1 | 08, 09 | Dfc (Cfb, Dsb, Dfb, Dwc) | 488 (55%) | 33% | 1894–2021 |
| TOTAL | 9.97 | 100 | | | | | |

[1] Based on Köppen Classification (Kpn) [15]. [2] as of February 2021. [3] Based on Water Survey of Canada (WSC) data by drainage region, of entire hydrometric data record (active + discontinued), number of gauges with "R" code.

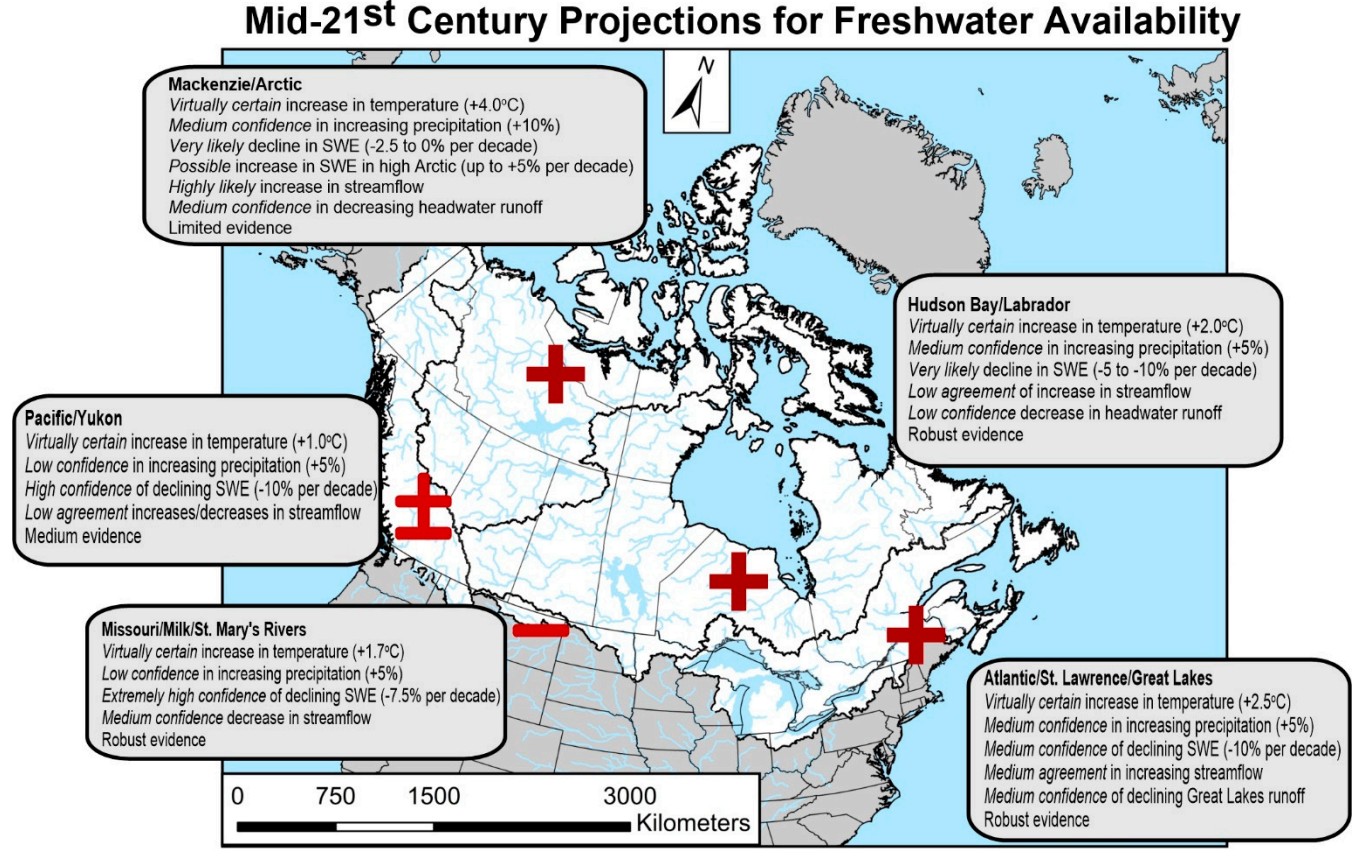

**Figure 4.** Projected change in streamflow for the mid-21st century in Canada's major drainage basins, and synthesis of projected hydroclimatic changes reported by CCCR2019 [3]. Degree of confidence determined by agreement among various studies and projections, while robustness refers to the amount of evidence available [17].

## 2.1. Emerging Continental Datasets

Crucial to the establishment of accurate continental-scale prediction systems are continuous and spatially congruent datasets for model forcing and evaluation. Several new

datasets have emerged recently that are important to highlight. The Canadian Precipitation Analysis (CaPA) reanalysis continues to be important for Canadian hydrologic analysis [18]. In the past, one of its drawbacks was the limited timeseries of data available, however, a Regional Deterministic Reanalysis Simulation (RDRS) is now available through the Canadian Surface Prediction Archive (CaSPAr) from 2000 to 2018, along with other versions of CaPA including a fine resolution (2.5 km) product [19]. The CaPA RDRS will soon be extended back to 1980 and made available through CaSPAr, making this highly suitable for forcing continental-scale Canadian models.

Global Water Futures (GWF) has made a strategic investment in advancing high-resolution forcing data for western Canada, to improve the spatial representation of precipitation over the Rocky Mountains. GWF-WRF (Weather Research and Forecasting model) provides 4 km gridded climate data over the historical period (2000–2015) and into the future (RCP 8.5) over western Canada [20]. There is also a 4 km implementation covering most of the continental United States, up to 73° N latitude, available from 1995 to 2015.

A recent addition to the reanalysis landscape is the serially complete data for North America (SCDNA) product that provides a 10 km reanalysis of precipitation and minimum and maximum temperature from 1979 to 2018 across all of North America [21]. Through the Canadian Centre for Climate Modelling and Analysis, there is improved dissemination of 10 km regridded and downscaled climate data for Canadian watersheds to standardize future analyses and improve intercomparison [22].

Emerging opportunities for more process-based model evaluation continue to be developed through the integration of remote sensing products into models, and via auxiliary data such as hydrologic tracers. Recently, a Canada-wide isotopes in streamflow dataset was released that provides stable isotopes of water in more than 300 rivers surveyed by Environment and Climate Change Canada [23,24], facilitating regional analyses of evapotranspiration partitioning and evaporation relative to inflow ratios.

### 2.2. Arctic and High-Latitude Drainage Regions

Greater than 60% of the water in Canada drains north, into and through high-latitude regions that contribute to the Arctic basin. Continental hydrologic prediction plays a particularly important role in high-latitude regions, not only because of the rapid changes being experienced, but owing to the particularly high percentage of ungauged landmass. Déry et al. (2016) noted statistically significant increases in northern river discharge across Canada using a gap-filled, outlet-corrected observational discharge timeseries [25], which corroborate findings reported by Durocher et al. [26]. This gap-filled timeseries has since been extended to 2018 to form an important observational record for assessing model predictions in Arctic-draining basins.

The role of freshwater discharge in determining ocean circulation and sea ice formation and breakup has garnered increasing attention from the research community [27]. Both the Mackenzie/Arctic and Hudson Bay systems provide freshwater input to the Arctic basin. Stadnyk et al. (2021) recently utilized the Hydrological Predictions for the Environment Arctic-HYPE model to map the impact of changing freshwater discharge across the entire pan-Arctic region, including the "Canada basin" of the Arctic. A potential increase of 22%—more than twice previous estimates based only on historic periods [26,28–31]—was projected when future periods were included in the analysis, generating a continuous 90-year timeseries of Arctic river discharge. Discharge from all Canadian basin rivers are projected to increase (statistically significant), however regulated rivers at a slower rate than non-regulated river regimes [2]. Among the 12 largest (by mean annual discharge volume) pan-Arctic rivers exhibiting the most significant rate of increase were the Mackenzie and Yukon Rivers in northern Canada.

### 2.3. Continental Interior and Hudson Bay

Several large, interdisciplinary projects across Canada in the past decade have contributed to the mapping of continental-scale freshwater discharge within Canada's interior

basins, including the BaySys project focused on Hudson Bay, Global Water Futures Integrated Modelling Programme for Canada (IMPC) focused on the Nelson River basin, and the recently started Nelson Multimodel Intercomparison Project comparing models on a process level under both regulation and future climates. Work within the FloodNet project has also targeted flow forecasting within the highly complex and poorly understood Prairie Pothole region of the Nelson River basin.

Under the BaySys project, a custom implementation of the Hydrological Predictions for the Environment (HYPE) model was established for the Hudson Bay basin that incorporated prairie potholes, frozen soils, lake parameterizations, and regulation including dams and diversions [32]. To improve model evaluation, gap-filled, outlet-based observational discharge data records were derived for all rivers entering Hudson Bay [25]. Future projections demonstrated greater sensitivity in the western tributaries to changes in future climate (warmer temperatures and higher precipitation) than the eastern (Quebec basin) tributaries resulting from the inflow-limited nature of the prairie drainage region and sensitivity to changes in evapotranspiration [8]. Similar findings have been reported in the literature for smaller, headwater basins of the Nelson River that are heavily influenced by the dynamic connectivity of the Prairie Pothole region [33–35].

The differences between the eastern slopes of the Canadian Rocky Mountain range (the headwaters of the Nelson River) and the Palliser Triangle region [36] of the prairies are notable. Significant increases in winter discharge are reported, with declining summer and late fall discharge by numerous studies [3]. The most significant declines in freshwater discharge across Canada were notably in the tributaries draining the eastern Rockies of western Canada, with many rivers across the Prairies indicating no significant trend as a result of evapotranspiration outcompeting increasing precipitation under warmer climates [3].

The eastern tributaries of Hudson Bay are projected to experience more significant increases in precipitation and, as reported by extensive modelling performed by Hydro-Québec, are expected to generate up to 15% more discharge into the future [37]. Similar modelling by Manitoba Hydro on the western tributaries suggests more modest increases in discharge for the Nelson River and its tributaries [38]. Overall projected increases in Hudson Bay discharge are anticipated to be up to 20% in some regions primarily due to increased winter flows, with significant uncertainty across the western portion of the basin [4].

### 2.4. Atlantic and Great Lakes

The northern regions of the Atlantic drainage area include parts of Labrador, including the Churchill River, which is of note because of its significance for hydropower production in the Atlantic Provinces. Mean annual runoff change between historic and future periods was assessed for Labrador by Roberts and Snelgrove (2015), who projected that runoff was increasing in Labrador's Churchill River under a series of regional climate model simulations [39].

In the central Great Lakes region, though hydrologic studies have been numerous over the past few decades, few have sought to examine freshwater availability across the entire region. The dominance of the Great Lakes themselves within this region adds a significant complicating factor, along with the extensive agriculture and irrigation practices, flow regulation, and large urban centres. The Great Lakes Regional Intercomparison Project (GRIP), under the IMPC project, is an initiative to develop and compare a variety of models ranging in structural and parameter complexity and originating from different modelling centres and groups. The project started by looking at the Lake Erie watershed for natural and regulated basins [40] and has now moved to setups that include the entire Great Lakes continental drainage region. In this region, spring peak discharge is generally anticipated to occur earlier and decrease across much of the Great Lakes and Atlantic basins, with some exceptions noted in Québec tributaries. Atlantic and Labrador drainage regions are anticipated to see an increase in future discharge, with the entire continental drainage basin

seeing increasing extreme summer rainfall events that are more severe and frequent [3], which presents a risk for increased flood events.

### 2.5. Pacific and West Coast

The limited number of studies at a continental-scale along the tributaries of the western Rocky mountain drainage, or Pacific drainage basins suggest both increases and decreases in discharge, with little to no consensus across the region but highly variable from one basin to the next [3,41–43]. This is not surprising given the complexity of mountain runoff and reliance on snowpacks and glaciers that are highly variable and influenced heavily by changing climatic conditions. While total annual runoff may remain relatively stable in a future, warmer climate, its timing will shift considerably from the warm to cool seasons [41,44]. Spring freshet is anticipated to occur earlier (up to one month) in the majority of these basins as a result of increasing temperatures and earlier snowmelt under future climates [45,46]. Continued glacier retreat in the 21st century will reduce their buffering capacity on summer low flows, particularly during warm, dry years [47]. A regional study of the western Canadian basins suggests increasing water surplus (precipitation minus potential evapotranspiration) along the Pacific coastline in all seasons but summer, where conditions are projected to be hotter and drier [48]. The loss of vast tracts of montane forests from timber harvesting, wildfires and pest outbreaks in a warming climate may also be altering surface hydrological processes including increased winter snow accumulations and earlier, faster spring melts [49]. Hydrological extremes may also be exacerbated in Pacific coastal watersheds such as the Fraser owing to projected increases in the frequency and intensity of landfalling atmospheric rivers, or narrow bands of concentrated moisture transport in the upper atmosphere [50].

### 3. The Role of Anthropogenic Regulation

Given the focus of our review is on freshwater availability, it is important to consider anthropogenic interventions impacting runoff and river networks enroute to continental outlets. Dams, reservoirs, river diversions, and irrigation withdrawals complicate freshwater prediction efforts due to the high degree of regulation within each drainage region (Table 1), which impacts both the timing and magnitude of discharge on annual and subannual timescales [25,51]. Studies have shown an increasing trend for river fragmentation and regulation over recent decades [52]. Within the Hudson Bay drainage region, >70% of the discharge entering Hudson Bay is considered regulated, with 47% considered to be intensely regulated [51,52]. If the effects of regulation are not considered, then prediction systems will ingest errors between observed and simulated flow into the model parameterization, which can significantly impact model robustness over long-term simulation and in future periods [53,54].

Hydroelectric regulation alters the timing and magnitude of ~50% of the freshwater discharge to Hudson Bay [25] and was therefore specifically targeted by the BaySys project [55] using scenarios that contrasted regulated and re-naturalized (pre 1970) states over both historic and future time periods. The findings demonstrate that while climate change is dominant on long timescales controlling inter-annual variability, intra-annual variability was predominately controlled by regulation and upstream storage availability [56]. Yassin et al. (2019) similarly studied the effects of reservoir regulation on river discharge across Prairie basins and developed a standalone model for incorporating reservoir regulation into continental-scale predictions [57]. Initiatives to incorporate river regulation, dams, diversions and irrigation withdrawals into network-based routing products play an important role for improving our understanding of the simultaneous impact of climate change and anthropogenic controls, and have the power to offer insights at the global scale [58]. New insights and modelling tools are the focus of the water resource management core modelling team under the GWF umbrella.

Observational data show that increasing river regulation in Canadian rivers over time has resulted in a distinct flattening of the average annual hydrograph, reducing

the seasonal cyclicity of river discharge signals [51]. Compounding this effect in some Arctic-draining rivers is climate change, which is increasing winter discharge and reducing summer and fall discharge signals, also leading to a natural flattening trend [2]. In other basins, however, climate change and regulation may offset each other resulting in insignificant or undetectable trends over time. Understanding the relative contributions of climate change and regulation on Canadian freshwater discharge is critical for future water supply availability. Implicit in this is also the role of human decisions affecting regulation, such as reservoir releases and the development of new infrastructure, and transboundary water share agreements. The IMPC project seeks to better understand the impact of decision-making and its integration with socio-economics and ecohydrologic needs [59]. A new water resources management model for the Nelson River basin extending from the Bow River headwaters downstream to the lower Nelson River outlet into Hudson Bay is fundamental to sustainable decision-making and evaluation of water allocations across the Canadian Prairies [60,61].

## 4. A Path Forward: Future Research and Modelling Needs

Given the importance of continental-scale prediction for climate change mitigation and adaption planning across Canada, there are several areas of research that must be advanced to support accurate prediction of Canada's freshwater supply.

The process of synthesizing projected changes in freshwater availability for this review proved to be difficult resulting from a lack of:

1. consistency in a historical baseline period from which to measure change,
2. consistency in (model) study design, and
3. observational data in some regions for model evaluation

It is for these reasons that instead a qualitative assessment was performed for freshwater availability, reporting only increasing or decreasing trends (rather than percent change). Furthermore, it is important to contextualize the confidence in reported projections using terminology established for the IPCC AR5 working groups given a lack of data-driven evidence in some regions (e.g., Arctic region), or considerable disagreement among modelled projections (e.g., western Hudson Bay region). A priority for the Government of Canada must be the unification of existing knowledge through the development of national data repositories, guidance on climate change impact assessment (including pre-determined historical baseline periods), and standardized methods for model benchmarking to ensure continuity across studies.

Owing to the expansive domain, diversity, and complexity of hydrological response to changing climates within the Canadian continental domain, we must leverage all available data resources for accurate evaluation of prediction systems. This includes meteorological forcing data and streamflow but requires particular emphasis on products that can be leveraged for process-based evaluation. There is an important distinction to be made between model accuracy and fidelity [62] that becomes increasingly important for long time scales and under climate change [54]. It is not only important to invest in data collection, but also model development to facilitate multi-objective optimization and decision-making utilizing these additional data.

To that end, what is needed at the continental scale are models that produce system-wide integrated outputs (e.g., streamflow simultaneous with water temperature and water quality). We must consider the non-stationarity imposed by climate change on all aspects of the system, including landcover and vegetation changes, necessitating models that can integrate dynamic vegetation change. This is critically important for the western Pacific basin, mountainous headwaters of the Prairie basins, and the Arctic basins as tree lines push northward and vegetation changes as permafrost thaws. We must move beyond stand-alone simulations of streamflow or other variables and take a much broader view of cumulative effects and propagation of change from upstream to downstream and across different ecosystems. This includes integration of dynamic climate-driven changes in freshwater discharge into ocean models, water quality models, and trophic structure mod-

els. Modelling systems that integrate river fragmentation and regulation are valuable for assessing the socio-economic and environmental impacts of anthropogenic alteration at the continental scale, which is crucial for sustainable water resource planning and management under climate change. Initiatives to improve the computational efficiency and capacity for integrated water resources management models are of paramount importance for understanding critical pathways and tipping points for decision-making on water management, and the cumulative impact on ecohydrology. Accurate depictions of reservoir operations, withdrawals, diversions, and water licensing are also critical to ensure decisions and future water scenarios accurately reflect supply and operations. With advances in computing sciences, fully dynamic and integrated models are increasingly possible across larger domains, such as the pan-Arctic and globally. Such integrated prediction systems offer significant potential to support operational decisions across Canada. For example, a model that integrates water quantity (streamflow) with temperature and river ice simulations could be used to provide operational information for ice road seasons in Canada's north, which are critical for northern transportation. Similarly, sea ice models that account for dynamic freshwater discharge and temperature will be better equipped to provide accurate information for northern shipping routes and ice breakup. Both the above examples are also important from a policy perspective as Canada considers Arctic security and sovereignty. Such endeavours require much better integration of academic science and inter-governmental operations, however, as well as investment in truly transdisciplinary science. For example, fish health and population density are intrinsically linked to the physical properties of streamflow (quantity, quality, and temperature), but with an absence of operational data to assess such linkages, biologists cannot establish environmental indicators or thresholds for sustainably managing fish population and health.

Integrated modelling has a significant role to play in supporting Canada's First Nations with their independence and community health and well-being. Clean water supply remains perilous for most communities [63–65], with a commitment from the Federal government to address this as a priority issue under Canada's commitment to Truth and Reconciliation. The problem, however, is complex and requires water supply and quality information for the planning and design of water treatment facilities in regions that are critically under-gauged. Integrated knowledge is also key here—with many citizen science and community-driven observation networks popping up across Canada there exists considerable opportunity to leverage additional data for model evaluation and operational planning. Owing to issues with quality control and assurance, a lack of standardization and continuity, the scientific community has been slow to accept such data networks. With some support from the proposed Canada Water Agency to invest in training, support, quality control and data repositories, however, this could be a considerable opportunity for Canadian hydrology. Similar and significant trust issues exist with the integration of traditional knowledge from Canada's Indigenous Knowledge Keepers, as well as Indigenous participation in 'western science' [66,67], with several studies highlighting the power and value of integrating knowledge within the hydrologic context [68–70].

Investments in science communication of integrated model outcomes are equally, if not more so, critical to the uptake and impact of model outputs. With the recent rise of skepticism in science [71–73], we need to do better as a scientific community to re-establish public trust. That begins with communicating our findings in non-technical, highly visual and interactive ways. The Virtual Water Gallery project launched by GWF is an interactive way to communicate the science of climate change to a much broader audience, and the c3s initiative funded by the European Union paired web developers with scientists to visualize climate change data and scenarios [74]. The BaySys project partnered with Manitoba Tourism and Via Rail to produce an e-book and interactive train car that would provide educational opportunities for passengers en route to Churchill, Manitoba regarding the impacts of climate change in Canada's North. As a community, it is important to invest in such endeavours, and to celebrate scientific achievement by more than just peer-reviewed publications, recognizing that the true impact of our science extends well beyond

knowledge and that now—more than ever—knowledge translation and operational uptake are essential for the future of Canada's water, food and energy supply.

**Author Contributions:** Conceptualization, T.A.S., S.J.D.; methodology, T.A.S.; resources, T.A.S., S.J.D.; data curation, T.A.S.; writing—original draft preparation, T.A.S., S.J.D.; writing—review and editing, T.A.S., S.J.D.; visualization, T.A.S., S.J.D.; funding acquisition, T.A.S., S.J.D. All authors have read and agreed to the published version of the manuscript.

**Funding:** This research was funded by the Natural Sciences and Engineering Research Council of Canada's Canada Research Chairs (NSERC CRC) program and the University of Calgary and the Industrial Research Chair program to UNBC.

**Institutional Review Board Statement:** Not applicable.

**Informed Consent Statement:** Not applicable.

**Data Availability Statement:** Not applicable.

**Acknowledgments:** The authors acknowledge the contributions of Andrew Tefs who assisted in the preparation of figures and data tables summarizing Canada's major drainage basins. We thank Kerry Black for her editorial support and input on traditional knowledge. We also thank the two anonymous reviewers who provided feedback that helped to improve this manuscript.

**Conflicts of Interest:** The authors declare no conflict of interest.

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
