# Peer review of "Canadian Continental-Scale Hydrology under a Changing Climate: A Review"

_water, doi:10.3390/w13070906_

Round 1
Reviewer 1 Report
Dear Authors,
I am writing this to submit my comments on your research article with the following details.
Manuscript title: Canadian continental-scale hydrology under a changing climate: A Review
Manuscript Number: water-1157002
Journal Submitted: Water
General Comments:
The is an excellent review article written in a way that suits a continental-scale hydrological study. The authors examined the role of climate change on Canadian hydrology, and current status of continental-scale climate change, and the anticipated impacts on sustainable availability of freshwater under the increasing influence of anthropogenic regulation. This review also focused on the regional-scale prediction of operational strategy and longstanding resource planning and management in Canada under rapidly changing climatic conditions. This way, it is an exciting article with regional-level implications.
The title is concise and represents the study. The abstract is well-focussed and to the point. The opening highlights the CCCR 2019 warning from Canadian scientists on climate change. Then it follows with the attractive heading “Wet gets wetter, dry gets drier.” I see this paper constructed in a good way and explains the intended investigation. I have a few comments based on my assessment.
Figure 1. Readjust the placement of text inside the figure.
Figure 2. Remove the background lines and short half circles in the figure borders. Also, use black for legends and axes labels.
Figure 3. The caption must be revised.
Table 1. Please provide the complete forms of abbrev used.
The section “The Role of Anthropogenic Regulation” should be populated more with the main emphasis on anthropogenic interventions (disconnectivity, effluents etc.). Also, highlight the need for restoration, where required.
In conclusion, this is an exceptionally well-conducted review.
References:
Okay.
Author Response
Reviewer 1
Dear Authors,
I am writing this to submit my comments on your research article with the following details.
General Comments:
The is an excellent review article written in a way that suits a continental-scale hydrological study. The authors examined the role of climate change on Canadian hydrology, and current status of continental-scale climate change, and the anticipated impacts on sustainable availability of freshwater under the increasing influence of anthropogenic regulation. This review also focused on the regional-scale prediction of operational strategy and longstanding resource planning and management in Canada under rapidly changing climatic conditions. This way, it is an exciting article with regional-level implications.
The title is concise and represents the study. The abstract is well-focussed and to the point. The opening highlights the CCCR 2019 warning from Canadian scientists on climate change. Then it follows with the attractive heading “Wet gets wetter, dry gets drier.” I see this paper constructed in a good way and explains the intended investigation. I have a few comments based on my assessment.
Figure 1. Readjust the placement of text inside the figure.
We have positioned the text to be entirely within the pie sections, where possible. The Drought label overlaps slightly with the adjacent section, but we feel this is a better option than a leader line and label.
Figure 2. Remove the background lines and short half circles in the figure borders. Also, use black for legends and axes labels.
Thank you for these suggestions. We have incorporated these edits into Figure 2.
Figure 3. The caption must be revised.
Since the reviewer did not specifically suggest how to revise, we assume this comment was directed at adding more detail. Therefore we have expanded the caption for Figure 3, and make specific reference to Table 1.
Table 1. Please provide the complete forms of abbrev used.
We have added “Water Survey of Canada” in place of WSC. For ecozones, these are not abbreviations but rather the actual Kpn (Köppen) ecozone climate classification codes. We have referenced these appropriately for readers.
The section “The Role of Anthropogenic Regulation” should be populated more with the main emphasis on anthropogenic interventions (disconnectivity, effluents etc.). Also, highlight the need for restoration, where required.
We respect the reviewers desire to have this information, and we agree it is also important information. However, this is not commensurate with the intent of our review (i.e., continental scale freshwater availability analysis), hence why we focused on regulation measures that detain, retain or affect the timing and volume of runoff making it to the continental outlets. We prefer to keep this review strategically narrower and more focused, but to make this distinction clearer we have 1) added a definition of the type of regulation impacts we are looking at here, and 2) our rationale for why we are limiting to this type of regulation. Note that we do address fragmentation (i.e., disconnectivity) in the context of river networks [line 238 in Section 3].
In our “Paths forward” conclusion section, we have added a brief discussion highlighting the need to understand the impact of river fragmentation and regulation, and to consider the socioeconomics and environmental effects of anthropogenic alteration of our waterways.
In conclusion, this is an exceptionally well-conducted review.
Thank you so much for your time in reviewing our paper. We appreciate your suggestions and your support for publication of our review.
Reviewer 2 Report
On the whole, the results of this study are interesting and have a practical and scientific value. As a review study, the work is well structured.
In my opinion, the only significant drawback of the work is the lack of generalization of all the numerous regional information provided by the authors. The best option for such a generalization would be a final schematic map of Canada's regionalization (with a complete legend) regarding the upcoming climate and hydrology changes. This map will significantly improve the presentation of the results of the manuscript. Otherwise, hydro-climatic forecasts scattered across numerous Canadian regions are difficult to perceive, especially for non-residents of North America.
Small remarks:
1. Line 62. Please define "the Palliser triangle" for an international reader.
2. Table 1. According to this table, the total drainage area within Canada is 10.79 million square kilometers. However, Canada's area is 9,984,670 square kilometers. The difference is almost 800 thousand square kilometers. This is more than the area of Saskatchewan! Make corrections.
3. Lines 268-269. Please define "atmospheric rivers" for an international reader.
Author Response
Reviewer 2
On the whole, the results of this study are interesting and have a practical and scientific value. As a review study, the work is well structured.
Thank you, we appreciate your support.
In my opinion, the only significant drawback of the work is the lack of generalization of all the numerous regional information provided by the authors. The best option for such a generalization would be a final schematic map of Canada's regionalization (with a complete legend) regarding the upcoming climate and hydrology changes. This map will significantly improve the presentation of the results of the manuscript. Otherwise, hydro-climatic forecasts scattered across numerous Canadian regions are difficult to perceive, especially for non-residents of North America.
We thank reviewer 2 for this comment! We immediately liked this suggestion, and went to work in generating a figure that could convey a summary of the projected changes in 21st century freshwater availability (recognizing that the historic context is less important to the objectives of the paper, we decided to focus on the relative change from the historic period). We did not wish to duplicate efforts in the CCCR2019, which largely provides a detailed summary of changes in temperature, precipitation, streamflow and hydrologic process (e.g., snow water equivalent). But we also recognize that if a reader is not Canadian, they have probably not read the CCCR2019, nor are they likely to! Therefore we decided the best option was to attempt to provide a summary of the projected changes reported among the CCCR2019 chapters pertaining to freshwater availability, which would be useful even for Canadian readers.
This turned out to be a daunting task as a result of a lack of consistency in study periods and historical baselines (observed data), the lack of observational records in some regions, and disagreement among studies/modelled projections in other regions. Therefore, we opted for a more qualitative review, summarizing confidence in the projections we report using the IPCC AR5 terminology that accounts for knowledge robustness (number of studies, amount of data, etc.) and confidence in the reported projections (i.e., agreement). We chose not to report likelihoods as our review is qualitative in nature (i.e., we do not have access to all the original model output or projections), therefore it was impossible for us to define statistically meaningful percentiles.
Figure 4 summarizes the projected changes in streamflow (freshwater availability) as increasing (+), decreasing (-) or both increasing/decreasing (±) by major drainage basin. Within these regions, we report the evidence that leads up to this result, including the projected temperature precipitation and SWE changes (reporting quantitative lower ends of the projected ranges by region to give the readers a feel for the spatial variation in climate change across Canada), and the resulting changes in streamflow. Confidence in these projections is reported by region, as well as the data robustness.
Small remarks:
1. Line 62. Please define "the Palliser triangle" for an international reader.
We have added “a semi-arid region of the western Canadian Prairies”
Table 1. According to this table, the total drainage area within Canada is 10.79 million square kilometers. However, Canada's area is 9,984,670 square kilometers. The difference is almost 800 thousand square kilometers. This is more than the area of Saskatchewan! Make corrections.
Excellent catch! We thank the reviewer for their keen eye. We had used an R script to compute the areas by adding up the Water Survey of Canada gauging areas, but somehow, we double counted some areas. We have gone back to the drawing board and recomputed them, and double checked with the reported WSC sub-drainage (major) region areas reported by StatsCan (https://www.statcan.gc.ca/eng/subjects/standard/sdac/sdacinfo1). We have updated our table – both drainage areas and corresponding % of landmass.
Lines 268-269. Please define "atmospheric rivers" for an international reader.
We have added “or narrow bands of concentrated moisture transport in the atmosphere” to describe what an atmospheric river is. You may know it as tropical plumes or connections.
Round 2
Reviewer 2 Report
Dear authors,
Thank you for the corrections you made.